# Integrated Evaluation of Dual-Functional DPP-IV and ACE Inhibitory Effects of Peptides Derived from Sericin Hydrolysis and Their Stabilities during In Vitro-Simulated Gastrointestinal and Plasmin Digestions

**DOI:** 10.3390/foods11233931

**Published:** 2022-12-05

**Authors:** Papungkorn Sangsawad, Sasikan Katemala, Danou Pao, Saranya Suwanangul, Rachasit Jeencham, Manote Sutheerawattananonda

**Affiliations:** 1School of Animal Technology and Innovation, Institute of Agricultural Technology, Suranaree University of Technology, Nakhon Ratchasima 30000, Thailand; 2Department of Animal Science (Agricultural Bio-resources and Food), Faculty of Agriculture, Kasetsart University Kamphaeng Saen Campus, Nakhon Pathom 73140, Thailand; 3School of Food Technology, Institute of Agricultural Technology, Suranaree University of Technology, Nakhon Ratchasima 30000, Thailand; 4Program in Food Science and Technology, Faculty of Engineering and Agro-industry, Maejo University, Chiang Mai 50290, Thailand

**Keywords:** sericin, bioactive peptide, ACE, DPP-IV, gastrointestinal digestion

## Abstract

Sericin, a byproduct of the silk industry, is an underutilized protein derived from the yellow silk cocoon. This research aimed to produce and characterize the bioactive peptides from sericin using various enzymatic hydrolysis methods. Alcalase, papain, neutrase, and protease were tested under their respective digestion conditions. Among the enzymes tested, neutrase-catalyzed sericin into specific peptides with the strongest dipeptidyl peptidase IV (DPP-IV) and angiotensin-converting enzyme (ACE) inhibitory properties. The peptides were subjected to a simulated in vitro gastrointestinal (GI) digestion in order to determine their stability. The GI peptides that were produced by neutrase hydrolysis continued to have the highest DPP-IV and ACE inhibitory activities. The neutrase -digested peptides were then fractionated via ultrafiltration; the peptide fraction with a molecular weight <3 kDa (UF3) inhibited DPP-IV and ACE activities. After being subjected to in vitro blood plasma hydrolysis, the UF3 was slightly degraded but retained its bioactivity. As a result of these findings, sericin peptides can be utilized as novel dietary ingredients that may alleviate some metabolic syndromes via the dual inhibitory properties of DPP-IV and ACE.

## 1. Introduction

Silk sericin comprises 30% silk proteins and 18 amino acids, with hydroxyl, carboxyl, and amino groups on its side chains [1]. The silk industry discards a substantial amount of sericin, which is considered a waste product. Recently, it was discovered that silk sericin contains essential amino acids, and it is used as an ingredient in medical, pharmaceutical, cosmetic, and food products [2]. As a source of food protein, it has been used as a flavor and texture enhancer in porridge, as an amino acid supplement in beverages, as an edible antioxidant in fatty foods, as an anti-browning agent in various foods, as an enhancer for the absorption of minerals, and as a dietary supplement [3]. In addition, sericin has been incorporated into numerous bulk structures, micro-nano formulations, and other biomaterials for tissue engineering and regenerative biomaterials [2]. Therefore, the valorization of sericin reduces environmental concerns and has high scientific and economic value.

Protein hydrolysate and peptides have been generated from natural protein substances worldwide [4]. They are widely used in the healthcare industry and in contemporary medicine. Many bioactive peptides with DPP-IV- and ACE-inhibiting activities have been extensively researched in recent years due to their potential antidiabetic and antihypertensive properties, which can be further developed into food products for health benefits. Diabetes and hypertension are the two most common risk factors for atherosclerosis and cardiovascular disease [5]. Both diseases frequently co-occur, indicating a substantial overlap in their etiology and disease mechanisms. In this context, diabetes is one of the most prominent global health diseases [6] and is frequently accompanied by hypertension; thus, both conditions should be avoided. Type 2 diabetes is characterized by hyperglycemia that is caused by impaired insulin secretion due to DPP-IV hydrolysis of two incretin hormones, which enhances glucose-induced insulin secretion during meals [7]. The suppression of DPP-IV activity is expressed in several tissues (kidney, intestine, and prostate), and it is a potentially effective therapy for type 2 diabetes. In the hypertensive stage, when ACE catalyzes the conversion of the decapeptide angiotensin I into the octapeptide angiotensin II, a potent vasoconstrictor is responsible for the inactivation of the vasodilator bradykinin, resulting in high blood pressure [8]. In modern medicine, ACE inhibition has been used successfully to treat hypertension. However, synthetic DPP-IV and ACE inhibitors have some negative side effects; thus, natural alternatives are being investigated [9].

Sericin is a protein that is slowly digested in the gastrointestinal tract [10]. The functional and nutritional properties of natural sericin have been improved through enzymatic hydrolysis. Although the optimum hydrolysis condition and method must be considered, hydrolysis conditions and enzymes influence the physicochemical and bioactivity properties of sericin peptides, resulting in variations in molecular weight and diverse bioactive peptides. Digestibility greatly affects the biological activity of sericin-derived peptides. Sericin and sericin peptides have been reported for various biological functions, such as antioxidants, antibacterial agents, tyrosinase inhibitors, hypocholesterolemic dietary supplements, anticarcinogens, and neuroprotectors [11]. However, the development of the dual function of DPP-IV and ACE inhibitors from sericin peptides has not been reported.

In previous decades, we conducted extensive research into the production of sericin from the yellow cocoon and the characterization of its biological activity, which demonstrated an anticancer effect against colon cancer cells by inducing apoptosis in malignant cells [12], decreasing cholesterol solubility in lipid micelles, and inhibiting cholesterol absorption by intestinal cells [13]. Furthermore, sericin-derived oligopeptides (SDO) that are derived from protease hydrolysis (from Bacillus species, Sigma, St. Louis, MO, USA) improved both in vitro and in vivo natural killer activities [14], decreased in vivo blood cholesterol concentrations [15] and demonstrated no toxicity to either red or white blood cells in mice [16]. However, the DPP-IV and ACE inhibitory potentials of sericin-derived peptides digested by various food-grade proteases have not been thoroughly investigated and evaluated.

To estimate the physiological and biological effects on the in vivo efficacy of bioactive peptides, i.e., their stability or survival during GI digestion and/or when traveling in the bloodstream, effective bioactive peptides should be able to reach their target sites in active form after ingestion [17]. Few studies have investigated the degradation of bioactive peptides in the gastrointestinal tract and in the circulatory system. Usually, the bioavailability of peptides has been measured using the in vitro simulated GI digestive model of Minekus et al. [18] and the blood protease digestion test, such as plasmin. Although the health benefits of sericin and its peptides have been reported, there is a lack of information regarding the effects of different enzymatic hydrolysis methods on the bioactivities of sericin peptides. The goal of this study was to characterize sericin peptides that were derived from four different proteases and to investigate the possibility of discovering peptides with strong anti-DPP-IV and ACE activities, which are relayed to diabetic and hypotensive syndrome. The peptide will be an innovative dual-functional food with the potential for antidiabetic and hypotensive activities. In addition, the stability of sericin peptides after simulated GI and blood plasma protease digestion was examined to estimate their prospective bioaccessibility. These results could be useful for developing food supplements from sericin peptides.

## 2. Materials and Methods

### 2.1. Chemicals and Reagents

The food-grade enzymes papain, alcalase, and neutrase were purchased from Brenntag Ingredients (Thailand) Company Limited. Protease (≥16 U/g, *Bacillus* sp.; P5985), pepsin (800 units/mg solid, porcine gastric mucosa; P7000), pancreatin (5.99 U TAME/mg solid, porcine pancreas, 8 × UPS), 2,4,6-trinitrobenzene sulphonic acid (TNBS), DPP-IV, gly-pro-p-nitroanilide (GPN) hydrochloride, ACE from rabbit lungs, N-[3-(2-furyl)acryloyl]-phe-gly-gly (FAPGG), acetonitrile, and trifluoroacetic acid (TFA) were obtained from the Sigma-Aldrich Company (St. Louis, MO, USA). The other chemicals utilized in this study were of analytical quality.

### 2.2. Sericin Extraction and Proximate Chemical Analysis

Sericin was isolated from silk cocoons (*Bombyx mori*) using the methods that were described in WO2013/032411 A1. In brief, silk was extracted under high pressure and heat using deionized (DI) water as a solvent. The extracted solution was concentrated using an evaporator until the final solid content was 20%. The AOAC technique [19] was used to estimate the approximate composition of the extracted sericin, including total solids, protein, fat, and ash content. The solid recovery was calculated as follows:(1)$$\mathrm{Solid}\;\mathrm{recovery}\;(\%)=\frac{\mathrm{Solid}\;\mathrm{content}\;\mathrm{of}\;\mathrm{recovered}}{\mathrm{Total}\;\mathrm{solid}\;\mathrm{of}\;\mathrm{the}\;\mathrm{original}\;\mathrm{sample}}\;\times 100$$

### 2.3. Production of Sericin Peptides

Five grams of sericin protein were dissolved in one hundred milliliters of DI water in order to prepare a sericin solution for enzymatic hydrolysis. The selected proteases were then added at 5% (E/S) and incubated at the following optimum temperatures and pH levels: papain (pH 7.0, 65 °C), alcalase (pH 9.6, 50 °C), neutrase (pH 7.0, 50 °C,) and protease (pH 7.0, 37 °C). This was followed by shaking, 4 h of incubation, and pH adjustments every 2 h. Then, the enzyme was inactivated at 95 °C for 20 min, the pH was adjusted to 7.0, and then it was centrifuged at 10,000× g at 4 °C for 10 min. For further analysis, the supernatant was collected and frozen at −80 °C.

### 2.4. Peptide Content and Degree of Hydrolysis (DH)

The TNBS method was used to quantify the peptide content with α-amino groups, and the assay was followed by the Adler method and Nissen’s approach [20]. For the reaction mixture, 10 µL of protein hydrolysate samples, 100 µL of 0.2125 M phosphate buffer, pH 8.2, and 50 µL of 0.05% TNBS solution were all put into a 96-well microplate. L-leucine was used to express the α-amino acid. A microplate reader was used to measure the absorbance at 420 nm, at 45 °C for 30 min (Varioskan LUX, Thermo Scientific, Vantaa, Finland). The peptide recovery was calculated as follows:(2)$$\mathrm{Peptide}\;\mathrm{recovery}\;(\%)=\frac{\alpha -\mathrm{amino}\;\mathrm{content}\;\mathrm{of}\;\mathrm{recovered}\;}{\mathrm{Total}\;\alpha -\mathrm{amino}\;\mathrm{content}\;\mathrm{of}\;\mathrm{the}\;\mathrm{original}\;\mathrm{sample}}\;\times 100$$

DH is the proportion of peptide bonds cleaved. It was also determined using the TNBS method and calculated using the α-amino groups as follows:(3)$$\mathrm{DH}\;(\%)=\frac{\alpha A1-\alpha A0}{\alpha \mathrm{At}}\;\times 100$$
where αA0 and αA1 represent the α-amino acid contents after 0 and 4 h of hydrolysis, respectively, and αAt represents the total α-amino acid (100% hydrolyzed in 6 N HCl at 120 °C for 24 h).

### 2.5. Bioactivity Assays

#### 2.5.1. DPP-IV Inhibitory Activity

The inhibitory action of a peptide against human recombinant DPP-IV was evaluated using the fluorescent substrate GPN. The fluorescence signals induced by the release of the p-nitroanilide group were measured to monitor the reaction. The inhibition was measured using the Lacroix and Li-Chan technique [21]. Twenty microliters of the sample were mixed with ten microliters of DPP-IV (0.01 U/mL) and incubated at 37 °C for 5 min in a 96-well microplate. Afterward, 50 µL of 30 mM GPN was added and incubated (37 °C for 30 min) using a microplate reader (Varioskan LUX, Thermo Scientific, Vantaa, Finland). The absorbance was monitored at 405 nm. For the blank, DI water was used instead of sample solutions. The following equation was used to calculate DPP-IV activity:(4)$$\mathrm{DPP}-\mathrm{IV}\;\mathrm{inhibition}\;(\%)=\frac{\mathrm{Slope}\;\left(\mathrm{positive}\;\mathrm{control}\right)-\mathrm{Slope}\;\left(\mathrm{test}\;\mathrm{sample}\right)}{\mathrm{Slope}\;\left(\mathrm{positive}\;\mathrm{control}\right)}\;\times 100$$

#### 2.5.2. ACE Inhibitory Activity

This assay is based on a chemical reaction between ACE and the substrate FAPGG, in which ACE hydrolyzes FAPGG to form FAP and GG, resulting in a decrease in absorbance at 340 nm. The inhibition was determined following the protocol of Martinez-Villaluenga et al. [22]. The reaction mixture contained twenty microliters of sample and ten microliters of ACE (1 mU/mL), preincubated at 37 °C for 5 min. Following that, 80 microliters of 0.5 mM FAPGG were added and incubated for 30 min at 340 nm with a microplate reader (Varioskan LUX, Thermo Scientific, Vantaa, Finland). DI water substituted the sample as a positive control. The activity was represented as the reaction rate (ΔAbsorbance/min), and the inhibitory activity was estimated using the following formula:(5)$$\mathrm{ACE}\;\mathrm{inhibition}\;(\%)=\frac{\mathrm{Slope}\;\left(\mathrm{positive}\;\mathrm{control}\right)-\mathrm{Slope}\;\left(\mathrm{test}\;\mathrm{sample}\right)}{\mathrm{Slope}\;\left(\mathrm{positive}\;\mathrm{control}\right)}\;\times 100$$

### 2.6. Molecular Weight (MW) Distribution

Size exclusion chromatography was used to determine the MW distribution of the peptides with a method slightly modified by Sangsawad et al. [23]. The sample (100 µL) was injected into a Superdex Peptide 10/300 GL column connected with a Fast Protein Liquid Chromatograph (AKTA PURE, GE Healthcare, Uppsala, Sweden). The UV_215_ nm was used to monitor the elution of 30 mL (flow rate of 0.6 mL/min) of 30% acetonitrile with 0.1% TFA in DI water. The standard for determining MW used aprotinin, two peptides (AGNQVLNLQADLPK and MILLLFR), cytochrome C, Hippuryl L-histidyl-L-leucine, and tyrosine.

### 2.7. Simulated In Vitro GI Digestion

Proteins and bioactive peptides can be further hydrolyzed into smaller sizes before being absorbed through the GI tract, resulting in a change in their bioactivities. The DPP-IV and ACE inhibitory effects of yellow silk sericin and sericin-derived peptides were investigated before and after the in vitro GI digestion test, which followed the method of Minekus et al. [18]. A 2.5 mL (40 mg/mL) sericin or peptide sample was prepared, followed by a pH adjustment to 3.0 before adding 2.0 mL of simulated gastric fluid. For a final concentration of 2000 U/mL, 0.5 mL of pepsin solution (800 U/mg solid) and 75 M of CaCl_2_ were added. After two hours of incubation in a water bath (37 °C) with shaking at 150 rpm, the process was stopped by raising the pH to 7.0. The gastric digested sample (pH 7.0) was mixed with 5 mL of simulated intestinal fluid, then 0.5 mL of pancreatin (5.99 U TAME/mg solid) and CaCl_2_ (0.3 mM) were added to achieve 100 U/mL. The reaction was shaken for 2 h at 37 °C and 150 rpm, then heated for 15 min at 95 °C before cooling for 5 min in an ice bath. The supernatants were collected after being centrifuged for 10 min at 10,000× *g* at 4 °C.

The amounts of α-amino groups in the digests were used to calculate the peptide content using the TNBS method described in Section 2.4. The DPP-IV and ACE inhibitory activities of the samples (pre- and post-GI digestion) were examined, as described in Section 2.5. The selection criteria were based on the peptides with the highest bioactivities after 4 h of digestion.

### 2.8. Optimization of Substrate Concentration

The peptides produced by neutrase hydrolysis had the highest DPP-IV and ACE inhibitory activity of all tested sericin-derived peptides. Furthermore, the bioactivity was retained after GI digestion. As a result, this condition was chosen as a production condition for maximal yield and bioactivity. The different sericin concentrations (5, 10, and 15 g protein/100 mL) were subjected to neutrase hydrolysis at 5% E/S and 55 °C. The hydrolysis times ranged from 0 to 8 h (maintaining a constant pH by adding 1 M NaOH or 1 M HCl). After each hydrolysis period, the enzyme was inactivated by 20 min of heating in a water bath at 95 °C. The peptides were centrifuged at 10,000× *g* for 10 min at 4 °C. The supernatant was collected to calculate the solid recovery and peptide content. The DPP-IV and ACE inhibitory properties of the peptides were investigated.

### 2.9. Ultrafiltration (UF)-Based Separation

Due to their highest DPP-IV and ACE inhibition, the sericin-derived peptides that were produced by neutrase hydrolysis were chosen for fractionation by partial purification using UF. The sericin-derived peptides were separated by a UF membrane (Amicon; EMD Millipore Corporation, Billerica, MA, USA).

The supernatant (10 mL) of neutrase-digested sericin-derived peptides was subjected to a first membrane (molecular weight cut-off of 10 kDa) and centrifuged twice with DI water (5000× *g* at 4 °C) to obtain the retentate with peptide sizes greater than 10 kDa. The permeate was then put through a second membrane (molecular weight cut-off of 3 kDa) to obtain the permeate and the retentate with peptide sizes of <3 and 3–10 kDa, respectively. The collected sample from each UF fraction was calculated for peptide yield and determined for DPP-IV and ACE inhibitory activities.

### 2.10. Simulation of the Plasmin Hydrolysis

After being absorbed by the epithelial cells on the inner layer of the GI tract, peptides enter the human blood circulation system and undergo further enzymatic hydrolysis. Plasmin is a human plasma serine protease that can hydrolyze various blood plasma proteins. Bioactive peptides in the bloodstream are known to be hydrolyzed, resulting in changes in their sizes and bioactivity. Based on Section 2.9, the bioactive sericin-derived peptides with the highest DPP-IV and ACE inhibitory activities were discovered in the UF3 fraction (<3 kDa). This fraction was selected to assess its stability and bioactivity using an in vitro plasmin digestion test according to the method of Sangsawad, Roytrakul, and Yongsawatdigul [23], with some modifications. Fifty milligrams of the sample were dissolved in a one-milliliter buffer (0.15 mM NaCl, 0.01% Tween 80, pH 7.4). Then, 20 µL of plasmin was added to the mixture at a final concentration of 0.05 U/mL for 4 h at 37 °C. After 0, 0.5, 1, 2, and 4 h, the sample was withdrawn, and the enzyme reaction was terminated for 10 min at 95 °C. Determining the degree of plasmin hydrolysis and the concentration of α-amino groups was performed, as described in Section 2.4. The inhibitory actions of DPP-IV and ACE were tested, as described in Section 2.5.

### 2.11. Statistical Analysis

All of the data were collected in triplicate and analyzed with a one-way analysis of variance (ANOVA). The Duncan procedure was used to calculate significant differences between the means (*p* < 0.05) using SPSS 16.0 for Windows (SPSS Inc., Chicago, IL, USA).

## 3. Results and Discussion

### 3.1. Proximate Composition

The extracted sericin solution had a solid content of 20.08%, comprising 98% protein, 2% ash, and 0% lipids and carbohydrates (on a dry basis). According to research conducted by Sasaki et al. [24], food-grade sericin powder is composed of 99.0% protein and 0% fat (dry basis). Therefore, the high protein content of extracted sericin may serve as an excellent source for producing peptides.

### 3.2. Effect of Enzymatic Hydrolysis on the Production and Bioactivity of Sericin Peptides

#### 3.2.1. DH and Peptide Yield

DH refers to the proportion of enzymatically hydrolyzed peptide bonds relative to the number of peptide bonds in the material. If the DH is high, it indicates that protein was hydrolyzed into smaller peptides and that the level of free amino acids is high. The progression of sericin hydrolysis in terms of DH was observed. Four different commercial proteases were used to hydrolyze the sericin, including alcalase, papain, neutrase, and protease. These enzymes are commercially available food-grade proteases that can produce bioactive peptides. The DHs (%) of these sericin peptides digested by the four enzymes were 32.44, 13.07, 9.37, and 5.00, respectively, as shown in Figure 1A. According to Zhou et al. [25], alkaline proteases, such as alcalase, have greater proteolytic activity than acid or neutral proteases, such as pepsin or papain. The result is consistent with previous findings regarding peptides that were prepared from salmon frames [26]. This study indicated that alcalase has a high level of proteolytic activity during hydrolysis. In addition, a high solid recovery (80%) was observed in the hydrolysate prepared from the four commercial enzymes (*p* > 0.05, Figure 1B), indicating that all commercial enzymes can hydrolyze the extracted sericin into peptides with high solubility.

#### 3.2.2. Bioactivity of Sericin Peptides

As shown in Figure 2A, all sericin peptide treatments inhibited DPP-IV, indicating that hydrolytic enzymes affected the DPP-IV inhibitory activity of sericin peptides. Interestingly, peptides that were derived from neutral protease at pH 7 (papain, neutrase, and protease) exhibited higher activity than alkaline protease (alcalase). Sericin protein (raw material) also showed activity. However, its activity was lower than neutrase hydrolysis. The DPP-IV inhibitory activity of neutrase-derived sericin peptides was 44.35%, indicating that enzymatic hydrolysis with neutrase had the greatest impact on increasing the bioactivity of sericin peptides. Compared to previous studies, sericin peptides that were derived from neutrase hydrolysis had higher DPP-IV inhibitory activity than wheat gluten, camel milk, Atlantic salmon, and pea protein hydrolysates [27]. Moreover, results indicated that DPP-IV inhibitory activity is not directly proportional to the degree of hydrolysis, which was found to be greatest in sericin peptides that were prepared with alcalase (Figure 1A). Thus, these results indicated that the bioactive sequence of peptides derived from sericin hydrolysis might inhibit DPP-IV more effectively than DH.

Many enzymes can produce peptide fragments during the enzymatic breakdown of dietary proteins. It is reasonable to speculate that peptides may have multifunctional properties. As a result, the peptides were evaluated for their potential hypotensive effect on ACE inhibitory activity. All sericin peptides derived from enzymatic hydrolysis could inhibit ACE more than sericin (Figure 2B). It is hypothesized that this activity was enhanced through enzymatic hydrolysis. The ACE inhibitory potential of sericin peptides derived from alcalase, papain, and neutrase exhibited the most potent inhibitory activity (58–62%) with insignificant variations. In comparison, sericin peptides hydrolyzed by protease demonstrated lower activity. Quist et al. [28] reported that an increase in DH led to a reduction in the size of peptides, thereby increasing the potential for ACE inhibition. However, the results of our study showed that the ACE inhibitory potential of sericin peptides was not correlated with their DH (Figure 1A). Compared to previous studies, sericin peptides derived from alcalase, papain, and neutrase hydrolysis had more potent ACE inhibitory activity than the date seed, Camelia Oleifera Abel seed, Tenebrio Molitor larva, and *Bombyx mori* larva hydrolysates [29].

Therefore, these results reveal that the sericin peptides obtained from neutrase hydrolysis provided the greatest number of peptides with dual functions to inhibit DPP-IV and ACE.

#### 3.2.3. Molecular Weight Distribution of Sericin Peptides

The molecular weight profiles of sericin and sericin peptides derived from a Superdex peptide column are depicted in Figure 3. The sericin possessed a molecular weight distribution of >10 to 5 kDa. Consistent with previous reports by Zhang et al. [30], sericin contains large-sized proteins, with a major peak estimated to be 10 kDa. According to Kundu et al. [31], sericin is produced in the middle portion of the silk gland and consists of polypeptides ranging in size from 24 to 400 kDa. Regarding sericin peptide, the molecular size of >10 kDa disappeared upon hydrolysis instead of becoming smaller. The molecular weights of sericin peptides produced with papain, neutrase, and protease were evenly distributed between 0.2 and 10 kDa. The major peak was composed of very short peptides, with molecular weights ranging from 1 to 5 kDa, which may contribute to its potent inhibition of DPP-IV.

According to DH, the molecular weight profile of peptides from alcalase hydrolysis was between 0.2 and 5 kDa, with a major peak at 0.2 to 1 kDa, indicating that it contained many short peptide chains. Small molecules 0.2 to 1 kDa may contribute to the highest ACE inhibitory activity. Consequently, this result indicated that sericin peptides with a molecular weight of 1 to 5 and 0.2 to 1 kDa would be the most effective inhibitors of DPP-IV and ACE, respectively.

### 3.3. Stability of Sericin Peptides upon In Vitro GI Digestion

According to scientific reports, proteins and peptides may be susceptible to hydrolysis by intestinal proteases [18]. The contribution of peptides derived from digestive enzymes was disregarded or not clarified in most published studies. We investigated the stability and bioactivity of sericin peptides during in vitro GI digestion with pepsin and pancreatin. After four hours of peptic and pancreatic hydrolysis, neutrase-treated sericin peptides exhibited the most stable and potent DPP-IV inhibition (Figure 4A). In the case of ACE inhibition, the activity of all peptides decreased after GI digestion relative to the activity level before digestion (Figure 4B). These differences indicated that all peptides were modified during digestion, leading to different peptides having varied DPP-IV and ACE inhibitory activity. However, the GI peptide that was generated from neutrase hydrolysis displayed the highest DPP-IV and ACE inhibitory effects after 4 h. Furthermore, these activities were shown to be more potent than GI peptide from sericin (raw material), indicating that enzymatic hydrolysis and GI digestion could enhance DPP-IV and ACE inhibitory actions.

The in vitro GI digestion of bioactive peptides should be used as one of the criteria for optimizing the production of a protein hydrolysate, as the activities of the GI digest are more closely related to health. Thus, our interesting discovery highlighted the ability of peptides produced by neutrase hydrolysis to have antihypertensive and hypoglycemic effects that target DPP-IV and ACE activities after GI digestion. Its potential bioactivities were more likely to reach the target organ after administration. It was, therefore, suitable for multifunctional health-promoting peptides.

### 3.4. Effect of Substrate Concentration and Hydrolysis Time

Since the neutrase hydrolysis peptide exhibited the most potent bioactivity following GI digestion, it was chosen for production optimization. The influence of substrate level (5%, 10%, and 15%) and time (0–8 h) on the sericin hydrolysis by neutrase was determined. During hydrolysis, the peptide concentration was increased (Figure 5A). The slope of the peptide concentration displayed the highest at 15% substrate, indicating that the highest reaction rate was observed. The concentration of peptides in 15% substrate increased rapidly for 0–4 h before reaching a plateau, indicating that the reaction rate was done. As a result, the optimal substrate concentration for production was 15%, with a hydrolysis time of 4 h.

High solid recovery indicates that high peptide yield can be utilized. It was calculated based on the solid content of the soluble fraction, as shown in Figure 5B. The previous study revealed that sericin comprises approximately 53% amino acids with strong polar groups, hence increasing its solubility in water [13]. Thus, the sericin solution had a high solid recovery at 0 h due to its high protein solubility. Furthermore, the results in Figure 3 (black line) demonstrated that the extracted sericin ranged in size from >10 kDa to 1 kDa, indicating that it was partially hydrolyzed by thermal treatment.

During 0–1 h of hydrolysis, the solid recovery of treatments increased rapidly to >80% before reaching a plateau, indicating that a large amount of the sericin substrate had been hydrolyzed to become soluble peptides. However, the solid recovery showed no significant difference from 1 to 8 h.

The bioactivity of sericin peptides was investigated using the same volume. The results exhibited significance in the inhibitory properties of DPP-IV and ACE (Figure 5C,D). The pattern of DPP-IV inhibitory activity across the three conditions was similar. The activity decreased from 0 to 4 h of hydrolysis. It was predicted that enzymatic hydrolysis altered the peptide’s MW profile, thereby diminishing its ability to inhibit DPP-IV. However, the condition of 15% substrate showed the highest DPP-IV inhibition. In contrast, the ACE-inhibiting activity of peptides after hydrolysis increased between 0 and 4 h (Figure 5D); this indicated that the peptides had been modified and appeared to be more effective at inhibiting ACE. The condition of 15% substrate at 4 h of hydrolysis showed the highest ACE inhibition. Therefore, 4 h of neutrase hydrolysis of sericin (SN4) is the optimal condition for obtaining peptides with the highest potential for dual activities.

### 3.5. Separation of SN4 with UF

UF is a common method for concentration and purification since UF molecular weight cutoffs (MWCOs) fit the size range of proteins and their peptide fractions. Table 1 displays the UF fractions of SN4 peptides from the membranes with 3 and 10 kDa. The samples with two distinct MWCOs were separated into three fractions labeled >10 kDa (UF1), 3–10 kDa (UF2), and <3 kDa (UF3). After sequential UF fractionation, the highest peptide yield and peptide content for DPP-IV and ACE inhibition were found in fraction UF3. It indicated that lower molecular weight peptides were responsible for DPP-IV and ACE inhibitory activities. Consistent with another study on whey protein hydrolysate [32], the results suggest that the peptide fractions below 3 kDa inhibited DPP-IV and ACE activities with the greatest potency.

### 3.6. Stability of Peptide upon In Vitro Blood Plasma Hydrolysis

Peptides in the bloodstream can be further hydrolyzed by plasmin, a human plasma serine protease. Plasmin has the same specificity as trypsin in cleaving peptides from Lys’Xaa and Arg’Xaa linkages [33]. Before entering clinical trials, the stability of bioactive peptides against blood protease hydrolysis must be evaluated. For our study, after GI digestion, the sample contained GI buffer, which interfered with the assay of in vitro blood plasma hydrolysis. Thus, the bioactive peptides in UF3 (<3 kDa) that exhibited the strongest DPP-IV and ACE inhibitory activities were chosen. As hydrolysis time increased, the α-amino group of UF3 increased simultaneously with ACE inhibitory activity (Figure 6A,B). It was significantly higher within the first hour of digestion, indicating that plasmin hydrolyzed the peptides into smaller fragments with greater inhibitory activity. Similar findings were found in which the plasmin hydrolysis of chicken breast hydrolysate and a peptide of KPLLCS increased α-amino acid content and their ACE inhibition during 0–3 h of digestion [23]. In addition, IVY derived from wheat germ hydrolysate persists for up to 4 h in human plasma [34]. However, there was a slight reduction in DPP-IV inhibition. Based on our findings, it is possible to hypothesize that consuming the peptide from the UF3 fraction generates peptides that contribute to DPP-IV and ACE inhibitory activities in the blood circulation system and the target organ.

## 4. Conclusions

This study discovered that the optimum conditions for producing DPP-IV and ACE inhibitory peptides were 15% sericin substrate, 5% neutrase (E/S), and 4 h for hydrolysis. Upon GI digestion, the GI peptides derived from neutrase hydrolysis exhibited dual activities. After fractionation, the small molecule in the UF3 fraction was considered to be a critical peptide group that exhibited the highest peptide yield and bioactivity. Furthermore, the UF3 fraction was partially hydrolyzed by plasmin, which increased ACE inhibitory activity while marginally decreasing DPP-IV activity. We conclude that the peptides derived from neutrase hydrolysis could be a source of DPP-IV and ACE inhibitory peptides following GI and blood plasmin digestions. It can be utilized as a novel dietary ingredient and nutraceutical product with health benefits. However, additional research is required to identify the sequence of amino acids that is responsible for the observed activities and to conduct animal and human feeding experiments to investigate the therapeutic efficacy of peptides derived from neutrase hydrolysis.

## Figures and Tables

**Figure 1 foods-11-03931-f001:**
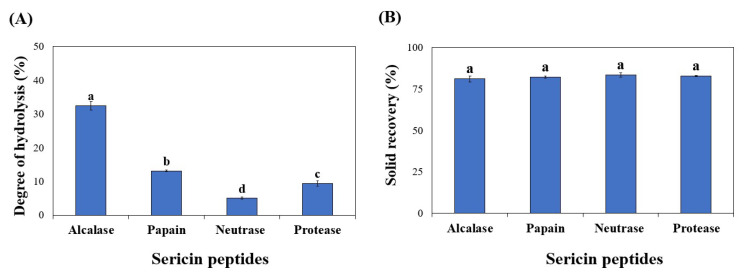
Degree of hydrolysis (**A**) and solid recovery (**B**) of sericin hydrolyzed for 4 h with different commercial enzymes. Significantly different values (*p* < 0.05) are represented by superscript letters (a–d).

**Figure 2 foods-11-03931-f002:**
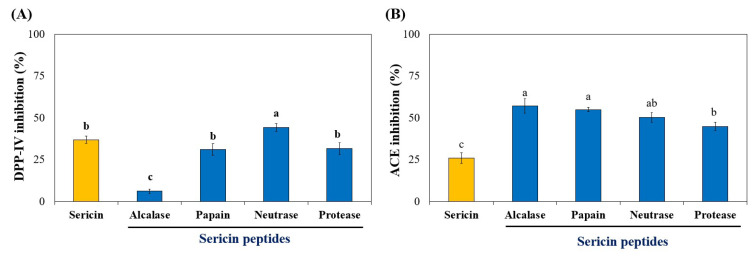
DPP-IV (**A**) and ACE (**B**) inhibitory activities of sericin hydrolysates. The DPP-IV and ACE inhibitory activities were measured in the reaction mixture at final concentrations of 2 mg and 0.2 mg Leu. eq./mL, respectively. Significantly different values (*p* < 0.05) are represented by superscript letters (a–c).

**Figure 3 foods-11-03931-f003:**
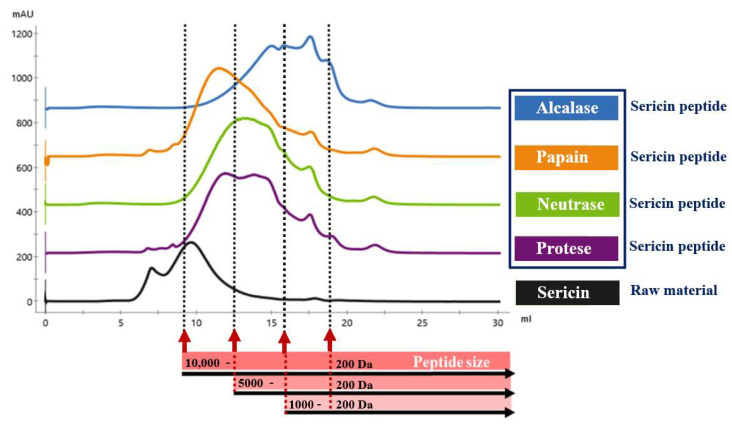
The chromatogram of sericin and sericin peptides on size exclusion chromatography.

**Figure 4 foods-11-03931-f004:**
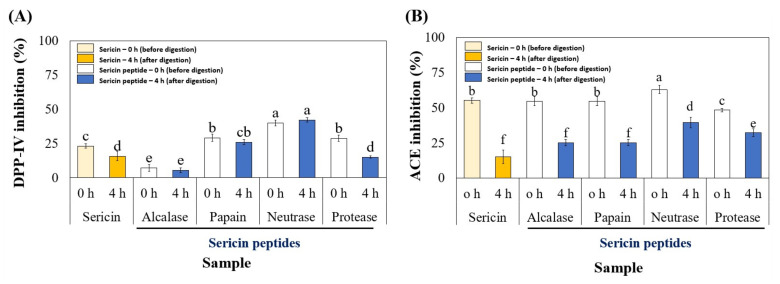
The DPP-IV (**A**) and ACE (**B**) inhibitory activities of sericin hydrolysates before and after 4 h of GI digestion. The DPP-IV and ACE inhibitory activities were measured in the reaction mixture at final concentrations of 2 mg and 0.2 mg Leu. eq./mL, respectively. Significantly different values (*p* < 0.05) are represented by superscript letters (a–f).

**Figure 5 foods-11-03931-f005:**
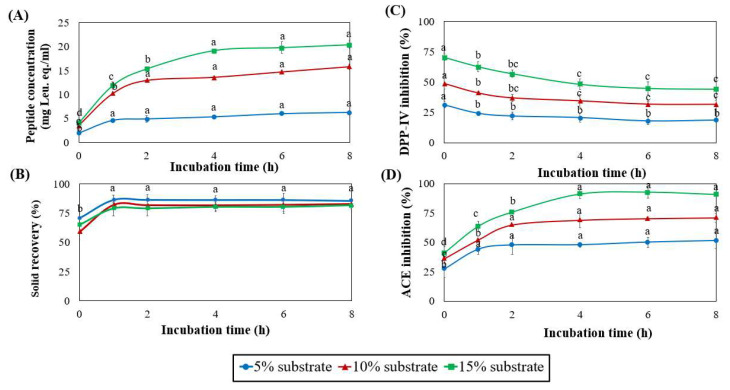
Peptide concentration (**A**), solid recovery (**B**), DPP-IV (**C**), and ACE (**D**) inhibitory activities of hydrolysate prepared from various sericin concentrations at 5, 10, and 15 g/100 mL, with 5% (E/S) of neutrase concentration, and a hydrolysis time of 0–8 h. The DPP-IV and ACE inhibitory activities were measured in the same dilution volume. Significantly different values within the line (*p* < 0.05) are represented by superscript letters (a–d).

**Figure 6 foods-11-03931-f006:**
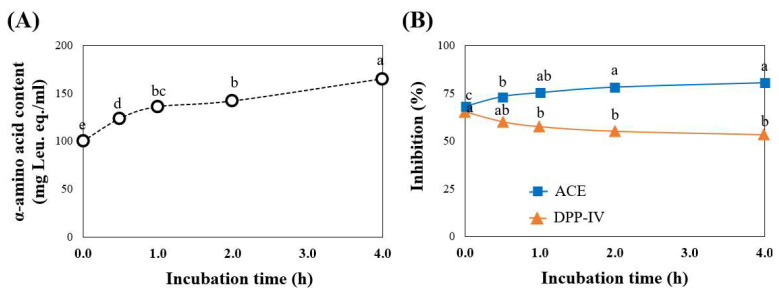
α-amino acid content (**A**), DPP-IV, and ACE inhibitor activities of UF3 incubated with human plasmin from different hydrolysis times (0–4 h) (**B**). The DPP-IV and ACE inhibitory activities were measured at final concentrations of 10 mg and 50 µg Leu equivalents/mL, respectively. Significantly different values within the line (*p* < 0.05) are represented by superscript letters (a–e).

**Table 1 foods-11-03931-t001:** DPP-IV and ACE inhibitory activities of sericin peptides derived from neutrase hydrolysis and peptide yields of ultrafiltration fractions.

Fraction	DPP-IV Inhibition *	ACE Inhibition *	Peptide Yield (%)
Crude	49.83 ± 1.59 ^b^	65.83 ± 1.59 ^b^	100.00
UF1	8.18 ± 5.93 ^d^	18.32 ± 2.34 ^d^	2.70 ± 1.12 ^c^
UF2	24.79 ± 6.65 ^c^	44.42 ± 4.52 ^c^	41.95 ± 0.02 ^b^
UF3	66.77 ± 4.79 ^a^	76.89 ± 2.64 ^a^	55.47 ± 5.57 ^a^

* The DPP-IV and ACE inhibitory activities were measured at final concentrations of 10 mg and 50 µg Leu equivalents/mL, respectively, in the reaction mixture. Significantly different values within the column (*p* < 0.05) are represented by superscript letters (a–d).

## Data Availability

The article contains the data and materials supporting the conclusions of this study.

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
