# Peer review of "Integrated Evaluation of Dual-Functional DPP-IV and ACE Inhibitory Effects of Peptides Derived from Sericin Hydrolysis and Their Stabilities during In Vitro-Simulated Gastrointestinal and Plasmin Digestions"

_foods, 2022, doi:10.3390/foods11233931_

Round 1
Reviewer 1 Report
Reviewers:
The manuscript “2004994” mainly develop “Enhancing Antidiabetic and Antihypertensive Properties of Sericin Peptides Derived after Simulated Gastrointestinal and Plasmin Digestions”. In my opinion, the intention of the experiment is reasonable, but the experimental design is relatively simple, and the experimental data are not enough to support the conclusion, so it needs modification.
1. In this manuscript, the author improved the activity of sericin by enzymatic hydrolysis, but the background of the article is too broad, so it is suggested to add control to reflect the better activity of sericin obtained after enzymatic hydrolysis.
2. The enzymatic hydrolysis process should take into account the influence of temperature.
3. In addition to animal experiments, cytotoxicity tests are also needed to verify the activity.
4. The principle of each detection method needs to be discussed.
5. The physicochemical indexes to verify the activity of hypertension and diabetes are too single.
Author Response
Dear Reviewer,
Thank you so much for your helpful comments and advice. We appreciate your time and efforts on this manuscript. We hope that our revised manuscript will meet the journal's requirements. The manuscript's topic was changed to "Integrated Evaluation of Dual-functional DPP-IV and ACE Inhibitory Effects of Peptides Derived from Sericin Hydrolysis, and Their Stability During Simulated Gastrointestinal and Plasmin Digestions". The major changes were changed in red, while the English changes were highlighted in yellow.
- In this manuscript, the author improved the activity of sericin by enzymatic hydrolysis, but the background of the article is too broad, so it is suggested to add control to reflect the better activity of sericin obtained after enzymatic hydrolysis.
= As a recommendation, information concerning the bioactivity of sericin (raw material) and its peptide produced following in vitro GI digestion was added as a control; Page 7, Lines 273-277 and Page 8, Lines 339-343.
Prior research in our group revealed that sericin-derived oligopeptides (SDO) or sericin peptide was generated from protease hydrolysis (from Bacillus species, Sigma, St. Louis, MO), which improved in vitro and in vivo activities such as natural killer activity, decreased blood cholesterol concentrations in vivo, and was non-toxic to red and white blood cells in mice. However, the enzyme used in previous studies was costly and produced a low peptide yield. To continue improving the potential of sericin-derived peptides and meet the demand for future pilot-scale production in terms of cost benefits, efficacy, and sustainability, food-grade proteases with similar activity or different modes of action must be investigated. The bioactivities of sericin peptides produced by various proteases, such as DPP-IV and ACE inhibitory activities, have not been studied or reported, particularly their dual activities. The information has been added to Pages 2-3, lines 78-105.
Therefore, this study aimed to create high-bioactivity of sericin peptides that would be an innovative dual-functional food against DPP-IV and ACE activity, which correlates to their potential for some antidiabetic and hypotensive activities. Furthermore, this study focused on the characterization of molecular weight distribution and partial purification. The stability of sericin peptides was investigated after simulated GI and blood plasma protease digestion. These discoveries could benefit in the study of animal and clinical applications in the future.
- The enzymatic hydrolysis process should take into account the influence of temperature.
= The hydrolysis conditions were carried out using the optimum conditions appropriate to each enzyme, as specified by specific recommendations, and the optimum conditions were provided with references.
Reference
Chen, J., Chen, Y., Xia, W., Xiong, Y. L., Ye, R., and Wang, H. (2016). Grass carp peptides hydrolyzed by the combination of Alcalase and Neutrase: Angiotensin‐I converting enzyme (ACE) inhibitory activity, antioxidant activities and physicochemical profiles. International journal of food science & technology, 51(2), 499-508.
Lapsongphon, N. and Yongsawatdigul, J. (2013). Production and purification of antioxidant peptides from a mungbean meal hydrolysate by Virgibacillus sp. SK37 proteinase. Food Chemistry, 141.2: 992-999.
Lapphanichayakool, P., Sutheerawattananonda, M., and Limpeanchob, N. (2017). Hypocholesterolemic effect of sericin-derived oligopeptides in high-cholesterol fed rats. Journal of natural medicines, 71(1), 208-215.
Wongngam, W., Mitani, T., Katayama, S., Nakamura, S., and Yongsawatdigul, J. (2020). Production and characterization of chicken blood hydrolysate with antihypertensive properties. Poultry Science, 99(10), 5163-5174.
- In addition to animal experiments, cytotoxicity tests are also needed to verify the activity.
= Our prior research had investigated the sericin and sericin-derived oligopeptides with in vitro and in vivo activities such as natural killer activity, lowered blood cholesterol concentrations in vivo, and non-toxicity to red and white blood cells in mice, etc.
We agreed with the recommendation; however, the experiment of this study was the first to investigate the condition for producing sericin peptides that exhibited high DPP-IV and ACE inhibitory activity. The experiment was designed as six experiments, as listed below. These studies demonstrated the technology for production, bioactivity, characterization with molecular size, and stability after GI and blood protease digestions. The next research will be examined upscaling, peptide sequencing, cell culturing, and in vivo experiments.
- Production of the bioactive peptide from sericin protein with different types of proteases; the peptides were released and exhibited different profiles and bioactivities.
- Characterization of sericin peptides correlated to their bioactivities with molecular weight distribution.
- The stability test of the peptides under in vitro GI digestion is important because GI enzymes can further hydrolyze peptides.
- Optimization for sericin peptide production from Neutrase hydrolysis by substrate concentration and time variation.
- Ultrafiltration method was used to characterize and investigate the effect of peptide size on their bioactivities and peptide yield.
- Stability of the highest bioactivity from the UF-fraction with the peptide size below 3 kDa under in vitro blood plasma protease hydrolysis.
- The principle of each detection method needs to be discussed.
= The information was added per suggestion, for example "Peptide content and degree of hydrolysis" (Page 3, Line 133), "Bioactivity assays" with DPP-IV and ACE inhibitory activity (Page 4, Line 147), "Simulated in vitro GI digestion" (Page 4, Line 178), "Ultrafiltration (UF)-based separation" (Page 5, Line 209), and "Simulation of the plasmin hydrolysis" (Page 5, Line 221).
- The physicochemical indexes to verify the activity of hypertension and diabetes are too single.
= This study aimed to discover and investigate the impact of food-derived bioactive peptides on human health. This research is the initial step in the "Production of Bioactive Peptides" using enzyme hydrolysis. Enzymatic hydrolysis provided the conditions for producing sericin peptides with the highest DPP-IV and ACE inhibitory activity. In modern medicine, the ACE inhibitor is derived from peptides, such as Teprotide (a nonapeptide isolated from the snake Bothrops jararaca); it has been used successfully to treat hypertension via an ACE inhibitory mechanism. Captopril is a commercial ACE inhibitor used to treat high blood pressure (hypertension). Commercial medicine based on DPP-IV inhibitors is produced from peptides such as Diprotin A (Ile-Pro-Ile) and Diprotin B. (Val-Pro-Leu). These are commercially available peptide products that inhibit ACE and DPP-IV.
The second step focused on "Oral Use of Bioactive Peptides" using in vitro GI and blood protease digestion model. These studies aim to monitor bioactivity after ingestion and absorption. It was determined to estimate their chance of survival inside the human body.
The next research will investigate cellular and animal models using the sericin peptide derived from neutrase hydrolysis.
Reviewer 2 Report
The manuscript entitled “Enhancing Antidiabetic and Antihypertensive Properties of Sericin Peptides Derived after Simulated Gastrointestinal and Plasmin Digestions” (Manuscript ID: foods-2004994) describes peptides obtained from sericin with DPP-IV and ACE inhibitory activity.
In my opinion the subject is very interesting and important but presentation and calculation the results must be rewritten before publication. Therefore, I think it needs analytical corrections.
Specific comments:
Line 22: It should be clarified. What does it mean: “5% (E/S) Neutrase, 15% sericin”? Was the amount of substrate 100 g or 15?
Line 21-23: It’s not clear. What does it mean SN4?
Line 65: If it is poorly soluble can it be used?
Line 65: What does it mean “intestinal protein”?
Line 99: What was the extraction method?
Section 2.2. In this subsection is terribly chaotic and there are many ambiguities. This subsection should be divided into individual analyses: protein determination and enzymatic hydrolysis.
Sections: 2.3.2 and 2.3.3. If the Authors use 20 µl of sample the results shouldn’t compare because the samples include different concentration of peptide.
Sections: 2.4 and 2.7: Please, split into two section: hydrolysis and peptide content analysis. After that should be provide section about The DPP-IV and ACE inhibitory activities methods.
Results and discussion
Lines 202-206 and 302-304: it is not clear. Methods for these analyses were not described.
Figure 4. Based on these results, it is not possible to determine which sample shows the best properties.
Figure 5. No markers on the axes.
Author Response
Dear Reviewer,
Thank you so much for your helpful comments and advice. We appreciate your time and efforts on this manuscript. We hope that our revised manuscript will meet the journal's requirements. The manuscript's topic was changed to "Integrated Evaluation of Dual-functional DPP-IV and ACE Inhibitory Effects of Peptides Derived from Sericin Hydrolysis, and Their Stability During Simulated Gastrointestinal and Plasmin Digestions". The major changes were changed in red, while the English changes were highlighted in yellow.
Specific comments:
Line 22: It should be clarified. What does it mean: "5% (E/S) Neutrase, 15% sericin"? Was the amount of substrate 100 g or 15?
= It means the sericin protein as a substrate concentration was 15%. The sentence was modified as " The condition for producing sericin peptide was 15% substrate (sericin), 5% neutrase (E/S), and 4 hours of hydrolysis time ", (Page 1, Lines 28-30).
Line 21-23: It's not clear. What does it mean SN4?
= It was defined as the optimum condition for the production of sericin peptide derived from Neutrase hydrolysis. The meaning as described, " The condition of 15% substrate at 4 hours of hydrolysis showed the highest ACE inhibition. Therefore, 4 hours of neutrase hydrolysis of sericin (SN4) is the optimal condition for obtaining peptides with the highest potential for dual activities." (Page 10, Lines 383-386).
Line 65: If it is poorly soluble can it be used?, What does it mean "intestinal protein"?
= The sentence was modified as " Sericin is a protein that is slowly digested in the gastrointestinal tract." (Page 2, Line 66). For more information, Sasaki et al. (2000) reported that sericin could be a useful agent treatment of constipation as a resistant protein. They recognized the preventive effect of dietary sericin against colon carcinogenesis in mice, similar to dietary fibre's effect. This result indicated that sericin is a resistant protein.
Furthermore, Lapphanichayakool et al. (2017) demonstrated that the sericin peptide extracted from thermal treatment and subjected to enzymatic hydrolysis could be decreased blood cholesterol concentrations in a rat model. It indicated that enzymatic hydrolysis improved the absorption of sericin peptides through the GI tract and into the target organ.
References
Lapphanichayakool, P., Sutheerawattananonda, M., & Limpeanchob, N. (2017). Hypocholesterolemic effect of sericin-derived oligopeptides in high-cholesterol fed rats. Journal of natural medicines, 71(1), 208-215.
Sasaki, M., Yamada, H., Kato, N. A. (2000). Resistant protein, sericin improves atropine-induced constipation in rats. Food Science and Technology Research, 6, 280-283.
Line 99: What was the extraction method?
= The experiment was added in "2.2. Sericin extraction of and proximate chemical analysis " (Page 3, Lines 114-120). " Sericin was isolated from silk cocoons (Bombyx mori) using the methods that were described in WO2013/032411 A1. In brief, silk was extracted under high pressure and heat, using deionized (DI) water as a solvent. The extracted solution was concentrated using an evaporator until the final solid content was 20%. The AOAC technique [19] was used to estimate the approximate composition of the extracted sericin, including total solids, protein, fat, and ash content".
Section 2.2. In this subsection is terribly chaotic and there are many ambiguities. This subsection should be divided into individual analyses: protein determination and enzymatic hydrolysis.
= We agreed with the Reviewer's suggestion. The subsection was modified and divided into;
2.1. Chemicals and reagents
2.2. Sericin extraction of and proximate chemical analysis
2.3. Production of sericin peptide.
2.4. Peptide content and degree of hydrolysis (DH).
2.5 Bioactivity assays
2.5.1. DPP-IV inhibitory activity
2.5.2. ACE inhibitory activity
2.6. Molecular weight (MW) distribution
2.7. Simulated in vitro GI digestion
2.8. Optimization of substrate concentration
2.9. Ultrafiltration (UF)-based separation
2.10. Simulation of the plasmin hydrolysis
Sections: 2.3.2 and 2.3.3. If the Authors use 20 µl of sample the results shouldn't compare because the samples include different concentration of peptide.
= The information was added per suggestion. It is unfair if we determine bioactivity with varying concentrations of peptides. The specific condition was added as follows:
- The experiment assay fixed the peptide concentration, as indicated in the captions for Figures 2, 4, 6, and Table 1. For example, in Figure 2, "The DPP-IV and ACE inhibitory activities were measured in the reaction mixture at final concentrations of 2 mg and 0.2 mg Leu. eq./ml, respectively." (Page 5, Lines 305-306)
- In Figure 5, the objective of this assay was to monitor their bioactivity during enzymatic hydrolysis, 0-8 hours. Thus, the experiment was performed using the same dilution volume. The sentence was added as "The DPP-IV and ACE inhibitory activities were measured in the same dilution volume." (Page 10, Lines 390-392)
Sections: 2.4 and 2.7: Please, split into two sections: hydrolysis and peptide content analysis. After that should be provide section about DPP-IV and ACE inhibitory activities methods.
= We agreed with the Reviewer's suggestion. The subsection was modified and divided into;
2.2. Sericin extraction of and proximate chemical analysis.
2.3. Production of sericin peptides.
2.4. Peptide content and degree of hydrolysis (DH).
2.5 Bioactivity assays
2.5.1. DPP-IV inhibitory activity
2.5.2. ACE inhibitory activity
Results and discussion
Lines 202-206 and 302-304: it is not clear. Methods for these analyses were not described.
= The information was added per suggestion. Protein content and proximate analysis were added in the section "2.2. Sericin extraction of and proximate chemical analysis" (Page 3, Line 114).
Figure 4. Based on these results, it is not possible to determine which sample shows the best properties.
= The sericin (raw material) and sericin peptides derived from enzymatic hydrolysis exhibited DPP-IV and ACE inhibition (Figure 2). In addition, the hydrolysates showed different molecular weight profiles (Figure 3). These results indicated that neutrase hydrolysis exhibited the most potential for both DPP-IV and ACE inhibition.
According to research data, proteins and peptides may be susceptible to breakdown by intestinal proteases. As a result, we studied the stability and bioactivity of sericin peptides after in vitro GI digestion. The activity following GI digestion is valuable information since it indicates that bioactivities are more likely to reach the target organ after ingestion.
Thus, the purpose of this section was to monitor the bioactivity before-after GI digestion. The selection criteria were focused on their bioactivity after GI digestion. For the results, the GI-peptide generated from Neutrase hydrolysis displayed the highest DPP-IV and ACE inhibitory efficacy after 4 hours. Furthermore, these activities were shown to be more potent than GI-peptide from sericin (raw material or starting material), indicating that neutrase hydrolysis and GI digestion could enhance DPP-IV and ACE inhibitory actions. Its potential bioactivities were more likely to reach the target organ after administration. It was, therefore, suitable for multifunctional health-promoting peptides.
Figure 5. No markers on the axes.
= The information was added per suggestion (Figure 6, Line 428).
Kind regards,
Papungkorn Sangsawad
Reviewer 3 Report
In general, it is an interesting work since it addresses health issues such as diabetes and hypertension. I am not an expert in the English language, but I recommend a thorough revision, since there are concepts that are wrong and I think it is the way of writing them.
In the abstract, the authors need to improve the information by highlighting the findings found in the work.
For example lines 24-25 what was the crucial compound? What was the purpose of following blood plasma hydrolysis by plasmin?
Introduction
Page 2. Line 65-66. Sericine is not a type of intestinal protein, because it is found on the fibres of raw silk.
Intruduction
In my opinion I believe that the auhors should delve a little deeper into what type of amino acids sericin has and therefore relate it to the possible antidiabetic and antihypertensive power.
Materials and Methods
Please explain hy the authors used alcalasa, papain, neutrasa in order to hydrolising sericin?
Page 4. Lines 170-176. It is said that the protein was hydrolyzed with alcalase, papain and neutrase and in M and M only hydrolysis with neutrase is described?
Page 5. line 188. What does the UF3 Fraction correspond to? Only this fraction was studied why?
Page 5. Line 188. Please explain how the authors determined the amount of total solids and how they related it to the chemical composition of sericin?. What type of amino acids make up this protein?, relate it to the literature.
Page 9. Line 312. The authors said the sericin has high water solubility, please explain, because I do not believe.
Conclusions
It seems to me that the conclusions are consistent with the results of the investigation.
Author Response
Dear Reviewer,
Thank you so much for your helpful comments and advice. We sincerely thank each Reviewer for their time and efforts on this manuscript and apologize for our English. We hope that our revised manuscript will meet the journal's requirements. The manuscript's topic was changed to "Integrated Evaluation of Dual-functional DPP-IV and ACE Inhibitory Effects of Peptides Derived from Sericin Hydrolysis, and Their Stability During Simulated Gastrointestinal and Plasmin Digestions". The major changes were highlighted in red, while the English changes were highlighted in yellow.
In general, it is an interesting work since it addresses health issues such as diabetes and hypertension. I am not an expert in the English language, but I recommend a thorough revision, since there are concepts that are wrong and I think it is the way of writing them.
= We agreed with the Reviewer's comment, and the manuscript was edited by MDPI Language Editing Services.
In the abstract, the authors need to improve the information by highlighting the findings found in the work.
= The information was added per suggestions.
- Neutrase-catalyzed sericin into specific peptides with the strongest DPP-IV and ACE inhibitory properties (Page 1, Line 21-22).
- The GI peptides that were produced by neutrase hydrolysis continued to have the highest DPP-IV and ACE inhibitory activities. (Page 1, Line 24-25).
- The neutrase -digested peptides were then fractionated via ultrafiltration; the peptide fraction with a molecular weight < 3 kDa (UF3) inhibited DPP-IV and ACE activities. (Page 1, Line 25-27).
- After being subjected to in vitro blood plasma hydrolysis, the UF3 was slightly degraded but retained its bioactivity. (Page 1, Line 27-28).
- Sericin peptides can be utilized as novel dietary ingredients that may alleviate some metabolic syndromes via the dual inhibitory properties of DPP-IV and ACE (Page 1, Line 28-30).
Introduction
Page 2. Line 65-66. Sericine is not a type of intestinal protein, because it is found on the fibres of raw silk.
= The sentence was modified as " Sericin is a protein that is slowly digested in the gastrointestinal tract." (Page 2, Line 66). For more information, Sasaki et al. (2000) reported that sericin could be a useful agent treatment of constipation as a resistant protein. They recognized the preventive effect of dietary sericin against colon carcinogenesis in mice, similar to dietary fibre's effect. This result indicated that sericin is a resistant protein.
Furthermore, Lapphanichayakool et al. (2017) demonstrated that the sericin peptide extracted from thermal treatment and subjected to enzymatic hydrolysis could be decreased blood cholesterol concentrations in a rat model. It indicated that enzymatic hydrolysis improved the absorption of sericin peptides through the GI tract and into the target organ.
References
Lapphanichayakool, P., Sutheerawattananonda, M., & Limpeanchob, N. (2017). Hypocholesterolemic effect of sericin-derived oligopeptides in high-cholesterol fed rats. Journal of natural medicines, 71(1), 208-215.
Sasaki, M., Yamada, H., Kato, N. A. (2000). Resistant protein, sericin improves atropine-induced constipation in rats. Food Science and Technology Research, 6, 280-283.
Line 99: What was the extraction method?
= The experiment was added in "2.2. Sericin extraction of and proximate chemical analysis " (Page 3, Lines 114-120). " Sericin was isolated from silk cocoons (Bombyx mori) using the methods that were described in WO2013/032411 A1. In brief, silk was extracted under high pressure and heat, using deionized (DI) water as a solvent. The extracted solution was concentrated using an evaporator until the final solid content was 20%. The AOAC technique [19] was used to estimate the approximate composition of the extracted sericin, including total solids, protein, fat, and ash content".
Introduction
In my opinion, I believe that the authors should delve a little deeper into what type of amino acids sericin has and therefore relate it to the possible antidiabetic and antihypertensive power.
= We agreed with the suggestion. The outcomes of this study were focused on an in vitro investigation consisting of six tests, described below. These studies provided information about sericin peptide production, bioactivities, characterization with molecular size, and stability after GI and blood protease digestion. These results were a primary step for producing the most potent DPP-IV and ACE inhibitory peptides.
The next research will examine the amino acid composition and identify the peptide sequence that exhibited the highest potent bioactivity. Furthermore, molecular docking was used to estimate the preferred orientation, affinity, and interaction of a peptide in the binding sites of DPP-IV and ACE. Thus, amino acids with potential antidiabetic and antihypertensive properties will be reviewed and discussed.
- Production of the bioactive peptide from sericin protein with different types of proteases; the peptides were released and exhibited different profiles and bioactivities.
- Characterization of sericin peptides correlated to their bioactivities with molecular weight distribution.
- The stability test of the peptides under in vitro GI digestion is important because GI enzymes can further hydrolyze peptides.
- Optimization for sericin peptide production from Neutrase hydrolysis by substrate concentration and time variation.
- Ultrafiltration method was used to characterize and investigate the effect of peptide size on their bioactivities and peptide yield.
- Stability of the highest bioactivity from the UF-fraction with the peptide size below 3 kDa under in vitro blood plasma protease hydrolysis.
Materials and Methods
Please explain why the authors used alcalasa, papain, neutrasa in order to hydrolising sericin?
= Sericin peptide production requires a food-grade protease. Furthermore, these enzymes (alcalde, papain, and neutrase,) were commercially available and inexpensive. The active sites in these enzymes differed; for example, papain preferred amino acids with aromatic side chains, such as Phe and Tyr. Neutrase enables the hydrolysis of protein to produce peptides with C-terminals consisting of hydrophobic amino acids, for instance, tyrosine, tryptophan or phenylalanine. While Alcalase has specificity for Alcalase was observed to cleave peptide bonds on the carboxyl side of Glu, Met, Leu, Tyr, Lys, and Gln. Alcalase contains multiple cleaving sites; thus, producing the peptides with short sequences is helpful.
In addition, the optimal pH for the hydrolysis of Papain, Neutrase, and Protease is 7, so the final product will not interfere with sodium chloride (NaCl). This salt will be a limitation in food applications. Alcalase has several cleaving sites, so it is interesting to produce with short peptides.
Page 4. Lines 170-176. It is said that the protein was hydrolyzed with alcalase, Papain and neutrase and in M and M only hydrolysis with neutrase is described?
= Lines 170-176 was the "optimization of substrate concentration" experiment. This section only used neutrase to hydrolyze the sericin. Because the results in Sections 3.1.2 and 3.2 indicated that the sericin peptide produced from neutrase hydrolysis has the greatest DPP-IV and ACE inhibitory activity, the bioactivity was retained after GI digestion. As a result, this condition was chosen as the best manufacturing condition for maximal yield and bioactivity. The optimum condition was studied with various substrate concentrations and hydrolysis times. This information was added per the suggestion (Page 9, Lines 355-386)
Page 5. line 188. What does the UF3 Fraction correspond to?
= UF is a common method for concentration and purification, since UF molecular weight cutoffs (MWCOs) fit the size range of proteins and their peptide fractions. Table 1 displays the UF fractions of SN4 peptides from the membranes with 3 and 10 kDa. The samples with two distinct MWCOs were separated into three fractions labelled > 10 (UF1), 10-3 (UF2), and < 3 kDa (UF3). Thus, the UF3-fraction corresponded to the permeated peptide fraction from a UF membrane with MWCOs 3 kDa. This sample was collected and calculated for peptide yield. DPP-IV and ACE inhibitory activities were also examined.
Only this fraction was studied why?
= Before entering clinical trials, the stability of bioactive peptides against blood protease hydrolysis must be evaluated. For our study, after GI digestion, the sample contained GI buffer, which interfered with the assay of in vitro blood plasma hydrolysis. In addition, it is well known that the smaller peptide size is stable or resistant under enzymatic hydrolysis. Thus, the bioactive peptides in UF3 (< 3 kDa) that exhibited the strongest DPP-IV and ACE inhibitory activities, were chosen.
Page 5. Line 188. Please explain how the authors determined the amount of total solids and
= The calculation of solid recovery was added in the manuscript (Page 3, Lines 120-121), and it was calculated as follows:
Solid recovery (%) = x 100
How they related it to the chemical composition of sericin?.
= Total solid is the total material (compound) in the solution of extracted-sericin. The solid content represented 20.08 % wet basis. After calculation based on dry basic, it contains 98% protein, 2% ash, and 0% lipids and carbohydrates. Thus, the solid recovery after hydrolysis will be implied to the recovered material or protein. According to research conducted by Sasaki et al. [24], food-grade sericin powder is composed of 99.0% protein and 0% fat (dry basis). Therefore, the high protein content of extracted sericin may serve as an excellent source for producing peptides.
Reference
Sasaki, M., Yamada, H., & Kato, N. (2000). Consumption of silk protein, sericin elevates intestinal absorption of zinc, iron, magnesium and calcium in rats. Nutrition Research, 20(10), 1505-1511.
- What type of amino acids make up this protein?, relate it to the literature.
= Elzoghby et al. 2015 reported that silk sericin protein contains major amino acid groups like serine (40%), glycine (16%), glutamic acid, aspartic acid, threonine, and tyrosine. Extraction of sericin using different methods affects to vary in the properties and bioactivities of sericin peptide. Our study extracted the sericin from thermal treatment and subjected it to further enzymatic hydrolysis with their optimum conditions, showing various DPP-IV and ACE activities. Thus, it could be due to differences in the active sites of each enzyme, then producing the peptides with different molecular weights and composition of amino acids. It is well known that the bioactive peptide sequence is a crucial factor in revealing the activity. However, we will identify the sequence of amino acids responsible for the observed actions in the additional research.
Reference
Elzoghby, A.O.; Elgohary, M.M.; Kamel, N.M. Implications of Protein- and Peptide-Based Nanoparticles as Potential Vehicles for Anticancer Drugs. Protein and Peptide Nanoparticles for Drug Delivery, 2015, 169–221.
Page 9. Line 312. The authors said the sericin has high water solubility, please explain, because I do not believe.
= Padamwar et al. 2014 reported that the secondary structure of sericin is usually a random coil, but it can also be easily converted into a β-sheet conformation. The solubility of sericin in water decreases when the structures are transformed from a random coil into the β-sheet form, especially in low temperatures. Our study isolated sericin from silk cocoons (Bombyx mori) by mixing silk with deionized water under high pressure and heat. Thus, it could be a reason for the increase in the random coil structure and the effect of increasing the solubility of sericin after the extraction process.
In addition, it is well known that the native sericin is hard to dissolve in water after drying into powder, like in the case of beta-casein. This extracted sericin will be gel-like when the solid content is more than 20%. However, the results in Figure 3 (black line) revealed that the exstrated-sericin composed of the protein and peptide ranged >10 kDa to 1 kDa; this was evidence that it was partial hydrolysis by thermal treatment. Moreover, it comprises approximately 53% amino acids with strong polar groups, increasing its affinity to water [Ref]. Thus, in this study, the sericin solution had a high solid recovery at 0-hour due to its high protein solubility.
Reference
Keawkorn, W., Limpeanchob, N., Tiyaboonchai, W., Pongcharoen, S., and Sutheerawattananonda, M. (2013). The effect of dietary sericin on rats. Science Asia, 39(3), 252-256.
Padamwar, M.N. and Pawar, A.P. (2004). Silk sericin and its applications: A review. Journal of Scientific & Industrial Research, 63, 323-329.
Kind regards,
Papungkorn Sangsawad
Round 2
Reviewer 1 Report
Accept
Reviewer 2 Report
All of the reviewer's comments have been incorporated into the manuscript.